# Psychological Impact and Compliance with Staying at Home of the Public to COVID-19 Outbreak during Chinese Spring Festival

**DOI:** 10.3390/ijerph19020916

**Published:** 2022-01-14

**Authors:** Huiwen Xu, Lin Liu, Luming Zhao, En Takashi, Akio Kitayama, Yan Zou

**Affiliations:** 1School of Nursing and Public Health, Yangzhou University, Yangzhou 225009, China; 006721@yzu.edu.cn (H.X.); 007044@yzu.edu.cn (L.L.); 2Faculty of Nursing, Nagano College of Nursing, Komagane 399-4117, Japan; takashi@nagano-nurs.ac.jp (E.T.); akitayama@nagano-nurs.ac.jp (A.K.); 3HSBC Business School, Peking University, Beijing 518055, China; Lumingzhao@pku.edu.cn

**Keywords:** novel corona virus, psychological impact, behavioral change, anxiety

## Abstract

In December 2019, COVID-19 was reported in Wuhan, China. Most of the studies related to the psychological impact and compliance with staying at home due to COVID-19 focused on ten days or one month after the initial “stay-at-home” phase of the COVID-19 pandemic. The early psychological impact and behavior change to COVID-19 during the Chinese Spring Festival (the start time for recommendations to stay at home) is uncertain. In this study, people from 23 provinces in China were recruited to participate in an online survey, using Credamo. Psychological impact and compliance with staying at home were evaluated by a self-designed and validated questionnaire. The results indicated that anxiety was the most often reported feeling (mean: 3.69), followed by sadness (mean: 3.63). Participants employed in foreign-owned companies were most likely to express anxiety and sadness. Overall, 61.8% of participants reported hardly going out, whereas 2.4% said they frequently went out during the initial “stay-at-home” phase of the COVID-19 pandemic. Participants with higher levels of anxiety and sadness were most likely to stay at home against the spread of COVID-19, as were female gender. This survey is an important study of the first reaction to staying at home during the initial “stay-at-home” phase coinciding with Chinese Spring Festival. Our findings identified factors associated with higher level of psychological impact and better compliance with staying at home recommendations during Chinese Spring Festival. The findings can be used to formulate precaution interventions to improve the mental health of vulnerable groups and high uptake of policy during the COVID-19 epidemic.

## 1. Introduction

In December 2019, an outbreak of atypical pneumonia associated with the 2019 novel corona virus was reported in Wuhan, the capital of Hubei province in China [1]. The 2019 novel corona virus infected pneumonia (NCIP) has been shown to spread from person to person through droplets and contact, and some severe symptom patients with NCIP will die [2]. Since then, the spread of coronavirus disease 2019 (COVID-19) was accelerated to all 34 regions of China by 30 January 2020. On the same day, the World Health Organization (WHO) declared the COVID-19 outbreak a public health emergency of international concern [3].

The number of cases has increased rapidly given that *chunyun*, a 40-day period with numerous people using domestic flights and trains across China, due to the Spring Festival, began on 10 January 2020 [4]. A report from the Chinese Ministry of Transportation showed the total number of public transportation trips during this period in 2019 was 2.98 billion, a number equivalent to all Europeans, Oceanians, and Africans collectively traveling at once [5]. This migrant population has returned to their home, which may therefore have transmitted COVID-19 in their hometowns. To mitigate the spread of the virus, the Chinese government has progressively adopted a series of measures, and staying at home was a primary and important measure since 23–24 January 2020, the time of the Spring Festival. Equal to Christmas of the West in significance, the Spring Festival is the most important holiday in China. Two features distinguish it from the other festivals. One is seeing off the old year and greeting the new. The other is family reunion. Before the eve of the Spring Festival, every family member tries to get back home from every corner of the country. Children and grandchildren will come back to their parents’ or grandparents’ house on this occasion. On the first three days of the festival, people will visit their close relatives and best friends, exchanging greetings and presents, which is known as the “New Year’s Visit”. A family reunion can bring people joy and also contributes to their psychological well-being [6,7]. However, due to the outbreak of COVID-19, people were confined to their homes instead of coming back to hometown and going to “New Year’s Visit”. Since the contagion coincided with the Spring Festival holidays, the most important celebration of the year, the psychological impacts of separation at a time of family reunion are unknown.

Previous studies have found that the general population experienced emotional stress, such as anxiety, worry, panic, and fright confronted with COVID-19 [8,9,10]. Many stayed at home and socially isolated themselves to prevent being infected, leading to a “desperate plea” [11]. Based on our knowledge, most of the studies related to the psychological impact of the COVID-19 on the general public focused on ten days or one month after the initial “stay-at-home” phase of the COVID-19 pandemic, identifying the mental health of the general public. The Spring Festival in China usually has seven days off from New Year’s Eve to the sixth day of the first lunar month. In 2020, New Year’s Eve falls on January 23 and Spring Festival falls on January 24. The time point of investigations in the previous studies were not within the time range of the Spring Festival holiday. Currently, there is not enough reliable information on first reactions to staying at home and the psychological impact of the general public right from the initial “stay-at-home” phase of the COVID-19 pandemic, especially pertinent with the spring holiday and uncertainty about the outbreak of such unparalleled magnitude.

Therefore, this present study represented the first psychological impact and compliance with staying at home survey conducted in a large, demographically representative sample of the population of China within the very early “stay-at-home” phase of the COVID-19 pandemic (4–5 days after) during the Spring Festival period. This study aimed to assess the associations between psychological impact and compliance with staying at home at the initial stage of the “stay-at-home”.

## 2. Methods

### 2.1. Sample and Variable Selection

In total, 1075 people were investigated across the China in the descriptive cross-sectional study on 28 January 2020, the fifth day of the Spring Festival holiday. The study was approved by ethical committees of school of nursing, Yangzhou University, and informed consent was obtained with participants prior to participation. We recruited a non-probability sample of Chinese adults aged 18 and older from internet users who participate in the Credamo marketplace. Credamo, providing functions equivalent to Amazon Mechanical Turk, is a population of internet users who participate in surveys in return for small monetary compensation (30 cents in this survey) and has been shown to be a good reliable data source [12,13]. The questionnaire of psychological impact and behavior change toward COVID-19 were posted on the Credamo marketplace and were sent to participants’ WeChat (similar to “WhatsApp”) through the platform. All questionnaires were collected the next day. Of them, 45 were invalid. Thus, the number of participants included in the analysis was 1030.

### 2.2. Measurements

The questionnaire consisted of two parts: demographics and psychological impact and behavior change toward COVID-19. Demographic variables include gender, age, profession and place of current residence.

We developed the questionnaire concerning psychological impact and behavior change toward COVID-19 according to guidelines of psychological crisis intervention for COVID-19 pneumonia by the National Health Committee, People’s Republic of China [14]. The questionnaire had two dimensions, psychological impact and indoor activities time changes, respectively. The questionnaire had a total of 17 questions: 8 questions regarding the extent of psychological impact, 1 question regarding change in frequency of going out (B1) and 6 questions regarding changes in indoor activities time with a Cronbach’s alpha of 0.827 and content validity of 0.86.

The extent of psychological impact dimensions was assessed with eight questions: “Facing COVID-19, I was anxious/angry/terrified/upset/sad/calm/confident/excited”. The response options were 1 = not at all, 2 = a little bit, 3 = moderately, 4 = quite a bit, or 5 = extremely. The total psychological impact dimension score ranged from 8 to 40 with a lower score indicating a less psychological impact. The Cronbach’s alpha coefficient of the psychological impact was 0.845 in our sample, indicating acceptable internal consistency.

Participants also completed the second part of the q questionnaire investigating the impact of the COVID-19 pandemic on changes in indoor activities time (Cronbach’s alpha of 0.780). Compliance with staying at home against COVID-19 was assessed with a single claim “From the onset of COVID-19 to 27 January 2020, how often you went out”. The response options were 1 = hardly going out, 2 = occasionally going out, 3 = frequently going out, or 4 = always outside. The higher score indicated less compliance with staying at home. The content regarding changes in indoor activities time was assessed with six questions: “During the COVID-19 outbreak, compared to previous years, what’s the change in my time spent surfing Internet/sleeping/playing game and listening music/working and studying/chatting/playing mahjong and cards at home?”. The response options were 1 = significantly decrease, 2 = slightly decrease, 3 = remained unchanged, 4 = slightly increase, or 5 = significantly increase.

### 2.3. Statistical Analysis

Descriptive statistics were calculated for sociodemographic and other selected characteristics of the participants. Two sets of ordinal logistic regression were used to assess whether sociodemographic factors were associated with the level of anxiety and sadness, and also to assess whether the level of anxiety and sadness was associated with staying at home adjusting for gender, age and occupation. All tests were two-tailed, with a significance level of *p* < 0.05. Analyses were performed using the SPSS software version 24.0 (IBM, Armonk, NY, USA).

## 3. Results

1075 people participated in the survey, of which 45 questionnaires were invalid, leaving 1030 valid questionnaires. Overall, 80.78% were ranged from 20 to 40 years and 58.4% were male. The age distribution of the participants is as follows: 11.75% are less than 20 years old, 42.33% are 21–30 years old, 38.45% are 31–40 years old, 6.21% are 41–50 years old, and 1.26% are 51–60 years old. Participants distributed in seven regions: eastern China (26.41%), southern China (13.50%), central China (13.30%), northern China (29.51%), northwest China (3.20%), southwest China (5.34%) and northeast China (8.74%). As for the occupational distribution of participants, 21.84% are students, 42.52% are company staff, 3.4% are civil servants, 28.45% are business people and 3.79% are foreign-owned company staff. At the same time, participants scored on the psychological dimension as follows: anxious (3.70 ± 1.14), sad (3.63 ± 1.17), fear (3.29 ± 1.16), angry (3.05 ± 1.25), upset (3.29 ± 1.17), calm (2.75 ± 1.16), confident (3.25 ± 1.23) and excited (1.98 ± 1.22).

### 3.1. Associations between the Sociodemographic Characteristic and Psychological Impact

Table 1 showed the associations between participants’ characteristics and psychological impact. Participants aged 21 to 30 years had a higher probability to have higher levels of anxiety compared to participants aged 41–60 years (OR 1.66, 95% CI: 0.061 to 0.955, *p* = 0.026). Male gender was significantly associated with less level of sadness (OR 0.71, 95% CI: −0.570 to −0.117, *p* = 0.003). Interestingly, foreign-owned staff were more likely to have higher levels of anxiety and sadness.

### 3.2. Compliance with Staying at Home

Overall, 61.8% of participants reported hardly going out, whereas 2.4% said that they frequently went out during the outbreak of the infectious disease. Additionally, 35.2% expressed that they occasionally went out and only 0.6% of participants were always outside during COVID-19 pandemic

### 3.3. Predictors for Greater Adoption of Staying at Home against COVID-19

Table 2 demonstrated that participants who had higher levels of anxiety and sadness were most like to comply with staying at home against COVID-19. However, level of fear was not associated with likelihood of staying at home. Compared with men, women were much more likely to adopt a stay-at-home policy.

## 4. Discussion

Our findings suggest that with respect to the initial psychological impact and compliance with staying at home of the general public until 28 January 2020, just 4–5 days into the country’s recommendations to stay at home, anxiety and sadness were the most frequently reported feelings and most of the participants (61.8%) adopted the stay-at-home policy during the epidemic of COVID-19 coinciding with Spring Festival. More importantly, the findings demonstrate that effective precautionary measures against the infection (staying at home) were critically dependent on participants’ psychological response and gender.

One curious result from this study is that people employed in foreign-owned companies reported statistically higher level of anxiety and sadness. Like SARS, COVID-19 is featured with human-to-human transmission. Employees of foreign companies are more likely to travel around the world for business than those of local companies. In the course of a business trip, the increased time spent outside may lead to an increased risk of COVID-19 exposure. This may lead to higher levels of anxiety and sadness among foreign employees. Additionally, there were now varying numbers of confirmed cases of COVID-19 abroad in Korea, Japan, Malaysia, Vietnam, Singapore, Sri Lanka, Nepal, Germany, France, the USA, Canada, and Australia, which means a global outbreak of COVID-19, and then may have a huge impact on the international economic. Epidemics have impacts on the economy of the countries and individuals, and workers’ worries about delays in work time and subsequent deprivation of their anticipated income may account for their higher levels of anxiety and sadness [15].

Our study shows that women had greater compliance with staying at home and reported more sadness, and 21–30-year-old women were more anxious than 41–60-year-old women. In ancient Chinese traditional culture, there is the idea that men are superior to women. For a new thing, men usually have more of their own opinions than women and will not accept it easily. However, women are more likely to follow rules set by those in power, which may explain why women were more compliant with staying at home than men. In addition, Chinese women tend to be more sensitive than men, so they were more likely to feel sad in the face of such a severe epidemic [16]. As for the reason that women aged 20 to 30 were more anxious than those aged 41 to 60, it may be that women aged 20 to 30 are in the stage of job hunting or job promotion, and the sudden epidemic may have a great impact on their work.

The spread of infectious diseases and the effect of government intervention are affected by individual behavior changes [17]. Therefore, understanding the influence of behavior changes on epidemics can be vital to successfully halt the spread of COVID-19 [18]. Compliance with staying at home among our sample was predicted by female gender, anxiety and sadness. Similar studies have found that levels of anxiety were associated with higher uptake of precaution measures by participants [19,20,21]. A study from Teasdale, Yardley, Schlotz, and Michie demonstrated that intending to stay at home during the simulated epidemic was associated with fear under protection motivation theory [22]. The finding that the level of fear of COVID-19 was the only predictor of positive behavioral change (e.g., social distancing, improved hand hygiene) is consistent with a recent study [23]. One possible reason may be that at the time conducting the study, China is in an earlier phase of the COVID-19 pandemic and the outbreak remains concentrated in Wuhan, Hubei province, the epicenter of the outbreak, with only a small number of cases reported in other parts of the country. Another reason participants experienced less fear than anxiety and sadness may be that experiences related to SARS-CoV epidemic may have laid foundation for confidence and sense of control in COVID-19 [24].

Our study is subject to a number of weaknesses. The clear disadvantage of this study is a smaller sample of each province, although our online sampling strategy has the ability to quickly arrange a survey and thereby investigate responses in real-time. The reason for the small sample size may be that the duration of the survey was relatively short, only one day; the platform of recruitment was not attracting enough participants. Our sample shows a bias for elderly participants who use the internet less and participants living in the northwest district and southwest district of China. The biases clearly limit the generality of our results. It would be desirable to compare the public’s psychological responses and compliance with staying at home in other geographical areas that were similarly affected, such as Singapore, American, England, Korea and Canada. Longitudinal studies survey changes in levels of psychological impact during this pandemic should be needed. Another limitation is that the level of anxiety, sadness, and fear are based on personal feelings, not from assessment by mental health professionals. However, self-reporting was paramount during the COVID-19 pandemic. Finally, because the extent concerning psychological impact did not focus on the impact of staying at home on public psychology caused by COVID-19, the information regarding the total psychological impact is not so relevant. However, our survey was conducted at the time when the state issued the stay-at-home requirements, and this study should partially show the impact of staying at home on public psychology.

In summary, during the initial phase of the COVID-19 outbreak in China, the general public rated the highest level of anxiety, followed by sadness. More than half of the participants complied with the staying at home during the Spring Festival. Foreign-owned staff were associated with a greater psychological impact of the outbreak. People were more likely to stay at home when they felt greater anxiety and sadness during the initial “stay-at-home” phase of the COVID-19 pandemic. Our findings can be used to formulate psychological interventions and public health interventions to improve mental health and control of outbreak during the COVID-19 epidemic.

## Figures and Tables

**Table 1 ijerph-19-00916-t001:** Association between the variables of age, gender and occupation and psychological impact ^a^.

Variables	Anxiety			Sadness		
	Mean ± SD	Adjusted OR (95% CI)	*p*-Value	Mean ± SD	Adjusted OR(95% CI)	*p*-Value
Gender						
Male	3.71 ± 1.14	1.03 (−0.190, 0.264)	0.769	3.53 ± 1.20	0.71 (−0.570, −0.117)	0.003
Female	3.68 ± 1.14	Ref		3.77 ± 1.12	Ref	
Age group						
≤20	3.73 ± 1.09	1.23 (−0.400, 0.814)	0.504	3.37 ± 1.26	0.75 (−0.885, 0.319)	0.357
21–30	3.79 ± 1.11	1.66 (0.061, 0.955)	0.026	3.61 ± 1.16	0.98 (−0.471, 0.420)	0.912
31–40	3.63 ± 1.15	1.34 (−0.145, 0.736)	0.189	3.74 ± 1.13	1.15 (−0.298, 0.583)	0.526
41–60	3.42 ± 1.23	Ref		3.62 ± 1.21	Ref	
Occupation						
Foreign-owned staff	4.08 ± 0.87	Ref		4.13 ± 0.73	Ref	
Company staff	3.69 ± 1.16	0.59 (−1.131, 0.078)	0.092	3.69 ± 1.17	0.61 (−1.102, 0.105)	0.105
Government staff	3.37 ± 1.45	0.39 (−1.779, −0.114)	0.026	3.51 ± 1.38	0.48 (−1.572, 0.088)	0.080

^a^ multivariable logistic regression done by ordinal regression on SPSS 24.0.

**Table 2 ijerph-19-00916-t002:** Predictors for greater adoption of staying at home against COVID-19 ^a^.

Variables	Level of Compliance with Staying at Home		
	Mean ± SD	Adjusted OR (95% CI)	*p*-Value
Anxiety level			
1	3.62 ± 1.68	0.19 (−2.384, −0.923)	<0.001
2	4.23 ± 1.20	0.52 (−1.273, −0.020)	0.043
3	4.17 ± 0.94	0.43 (−1.305, −0.374)	<0.001
4	4.56 ± 0.70	0.84 (−0.550, 0.211)	0.382
5	4.69 ± 0.59	Ref	
Sadness level			
1	3.71 ± 1.72	0.25 (−2.043, −0.708)	<0.001
2	4.04 ± 1.19	0.24 (−1.948, −0.903)	<0.001
3	4.10 ± 0.92	0.21 (−1.987, −1.117)	<0.001
4	4.61 ± 0.63	0.56 (−0.985, −0.162)	0.006
5	4.78 ± 0.49	Ref	
Fear level			
1	3.88 ± 1.69	1.95 (−0.152, 1.490)	0.110
2	4.31 ± 0.98	1.41 (−0.258, 0.943)	0.264
3	4.33 ± 0.86	0.86 (−0.667, 0.365)	0.567
4	4.56 ± 0.69	0.83 (−0.675, 0.308)	0.465
5	4.71 ± 0.60	Ref	
Gender			
Male	4.34 ± 0.95	0.62 (−0.755, −0.208)	0.001
Female	4.54 ± 0.86	Ref	
Age group			
≤20	4.23 ± 1.01	0.58 (−1.259, 0.185)	0.145
21–30	4.43 ± 0.92	0.76 (−0.834, 0.289)	0.341
31–40	4.45 ± 0.88	0.71 (−0.899, 0.211)	0.224
41–60	4.55 ± 0.90	Ref	
Occupation			
Company staff	4.39 ± 0.99	0.80 (−0.964, 0.517)	0.554
Government staff	4.31 ± 1.31	0.88 (−1.136, 0.875)	0.799
Business person	4.53 ± 0.81	1.15 (−0.615, 0.900)	0.712
Student	4.33 ± 0.90	0.84 (−0.982, 0.628)	0.666
Foreign-owned staff	4.59 ± 0.72	Ref	

^a^ Multivariable logistic regression done by ordinal regression on SPSS 24.0.

## Data Availability

The data presented in this study are available on reasonable request from the corresponding author. The data are not publicly available due to privacy.

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
