# Peer review of "Psychological Impact and Compliance with Staying at Home of the Public to COVID-19 Outbreak during Chinese Spring Festival"

_ijerph, 2022, doi:10.3390/ijerph19020916_

Round 1

Reviewer 1 Report

To the authors

I have read revised version of the manuscript “Psychological impact and compliance with staying at home of the public to COVID-19 outbreak during Chinese spring festival”. It is now clearer but still requires improvements and clarifications.

Some more detailed observations/questions:

Abstract:

”Participants reported the highest level of anxiety (score: 3.69), followed by sadness (score 3.63).” => ‘The results indicated that anxiety was the most often reported feeling (mean: 3.69), followed by sadness (mean: 3.63)’.

It is not clear what is the base for the result “Participants employed in foreign-owned company were most likely to express anxiety and sadness”. (see also comment a bit later). Important in the abstract? OR including results of other statistically significant factors as well?

r 29: => …to staying at home recommendations during… OR omit ‘staying at home during’

r 30-33 => The findings can be used to formulate…

Introduction:

Description of the importance of social gathering of families and friends and mental meaning of it is now better connected to the Chinese spring festival.

Could you/is it relevant to call measures of Chinese Government as ‘recommendations to stay at home’ (as you write ‘there is not enough reliable information on first reactions to staying at home and the psychological impact of the general public right)?

In various part of the manuscript, you mention that the study was conducted ‘within the very early phases of the outbreak of COVID-19 (4-5 days after)‘/’ just 4-5 days into the country’s outbreak of COVID-19’. However, in the beginning of the introduction, you tie ‘outbreak’ with the beginning of the epidemic (December 2019). Thus, I am still a bit confused with what you mean with ‘outbreak’ and am not sure, if ‘outbreak’ is the relevant term to the time the study was conducted. Please, consider this. Anyway, your response to the second comment (in earlier phase) convinced me and explained why the time for the present study was really important. It would be good, if you could include some standpoints from your answer into the text, e.g., into Introduction or ‘Sample and variable selection’.

Sample and variable selection:

Modify ‘During the survey, a total of 1075 questionnaires were collected, of which 45 were invalid.’ => ‘Of them, 45 were invalid. Thus, the number of participants included in the analysis was 1030.’ and move it into the end of ‘sample and variable selection’.

Measurements:

Do not include the questionnaire in the manuscript. Express questions you used in the study in the text.

Add and modify e.g., ‘The extent of psychological impact dimensions was assessed with eight questions: “Facing COVID-19, I was anxious / angry / terrified / upset / sad / calm / confident / excited”. The response options were 1=not at all, 2=a little bit, 3=moderately, 4=quite a bit, or 5=extremely.’

The information regarding the total psychological impact is irrelevant in the present study.

Modify and add e.g., ‘Compliance with staying at home against COVID-19 was assessed with a single claim “From the onset of COVID-19 to January 27,2020, how often I went out”.

Statistical analysis:

Add ‘were’ into ‘…whether sociodemographic factors were associated…’

Specify analysis used. (Logistic regression refers to the analysis with dichotomized outcome.)

Results:

Begin the descriptive table with age and gender.

Add means and standard deviations of psychological impact into the results (maybe in the same descriptive table as demographics and shortly in the text) (they were earlier in the table which you decided to omit after the comments and your consideration).

Add ‘s’ into associations and characteristics in heading p6 r188-89.

P6, r 194 in the parenthesis: Should it be 95% CI?

Is there some mistake as you say in the text ‘foreign-owned COMPANY staff was more likely to have higher levels of anxiety and sadness’ but in the table, there is just company staff, government staff and no reference group? Omit ‘adjusted’ from the table, as all variables are of interest?

I prefer expressing proportions of frequency of going out just in the text (and maybe in the same descriptive table as demographics) (and would omit the figure).

In Table 4, there is something odd with the rows.

Add e.g., ‘However, level of fear was not associated with likelihood of staying at home’.

Discussion:

‘were the main psychological impact’ => ‘were the most frequently reported feelings.'

Discussion regarding the result indicating that people employed in foreign-owned companies reported statistically SIGNIFICANTLY higher levels of anxiety and sadness should be rewritten. Justified reasoning is that epidemic have impact on the economy of the countries and individuals, and that workers’ worriers about delays in work time and subsequent deprivation of their anticipated income may account for higher level of anxiety and sadness. However, it is not clear how ‘human-to-human transmission of the virus and the concern about virus exposure in public transportation when returning to work’ may explain this result specifically. The sentence ‘And there were now varying numbers of cases abroad in Korea, Japan…, which have a huge impact on the international economic’ should be written in more informative and understandable way.

Modify the sentence ‘The effects of infectious disease transmission and government interventions are subject to individual behavior changes’ in more understandable way. Write the finding of your own study in past tense. Change ‘similar researchers’ into ‘similar studies/research (?). Modify the sentence ‘Also consist with a recently report level of fear was the only predictor of positive behavior change () was fear of COVID-19’ (what do you exactly mean with this?). If better, change ‘at the time of writing’ => ‘at the time conducting the study’ (?). What do you mean with ‘without instill across large swathes of the population’? Later part of the last sentence => e.g., ‘Another reason for our finding suggesting that the participants experienced fear less frequently than anxiety and sadness may be that experiences related to SARS-COV epidemic may have laid foundation for confidence and sense of control in COVID-19’.

Have you any idea, why women had greater compliance with staying at home and reported more sadness, and 21-30-year-old more anxiety than 41-60-year-old? Consider if discussion of these results is relevant.

Author Response

This manuscript is a resubmission of an earlier submission. The following is a list of the peer review reports and author responses from that submission.

Round 1

Reviewer 1 Report

To the authors

I have read the manuscript “Psychological impact and compliance with staying at home of the public to COVID-19 outbreak during Chinese spring festival”. The topic is important and interesting. However, according to my mind, the manuscript requires thorough improvements. The thread/theme of the study in not clear. You should give better and stronger reasons for the topic/aim of the study. Theoretical background should be improved. Description of the study variables should be more precise, uniform, and be presented more logically. There is quite much descriptive statistics in the results compared to study aim. Discussion should be improved and deepened. Language should be checked.

Some more detailed observations/questions:

-Describing the Chinese spring festival is good, since it is not familiar for all people in other countries. However, the description could be condensed a bit and, on the other hand, connected to phenomenon and importance of social gathering of families and friends and mental meaning of it.

-I wonder how 4-5 days (time frame in your study) make difference compared to 10 days or one month (in earlier studies) concerning COVID-19 outbreak and psychological impact of it.

-Concerning aim of the study: Can ‘how public reacted to staying at home during the Spring Festival’ be a study question? You did not analyse it.

-Content of the paragraph ‘Sample and variable selection’ does not include anything of variable selection. Number of the participant should be expressed in this paragraph. I wonder if it is reasonable to say that the data of the study is demographically representative as the participants were internet users? How about representativeness of different age groups?

-Omit outcome from the headline ‘outcome measurements’ since all variables descripted are not outcomes.

-Sentence on rows 105-108 is better -> ‘We developed the questionnaire concerning…. according to the guidelines…’

-Describe measurements more precisely and uniformly, and by variable by variable (sum variables). I prefer to express questions in condensed way in the text instead of the supplement. Were positive emotions reverse coded? Since you used logistic regression how anxiety and sadness were categorized? How did you categorize ‘staying at home’?

-What is reasoning for expressing emotions according to participants living in various provinces of China? Of course, you are able to observe difference between participants living in different provinces. Did you test the difference? Still, what is the meaning (cf. thread of the study)? What is the connection between districts and provinces?

-I prefer not to say ‘participants reported the highest level of anxiety…’ but e.g., the mean score of anxiety was highest among the participants…

-There is too much descriptive results (cf. thread of the study, aim of the study).

-row 161: do not say ‘participants ages 21 to 30 were significantly associated with…’ but that participants aged 21 to 30 years had higher probability to have higher level of anxiety compared to participants aged 41-60 years.

-Is there any other meaning of expressing changed time of indoor activities than descriptive statistics? Correctness of headline 3.3.?

-I did not catch the discussion concerning people employed in foreign-owned companies.

-Some contents of the paragraph beginning from row 240 could be removed to introduction (where logical) and some omitted.

Reviewer 2 Report

The sample size was too low for a vast country like China. The effect was "thinned" out by the seven regions involved. This meant that only just over one hundred cases were studied in each region, making the effect and claim of regional representation or generalisation even more difficult. It appeared likely that the survey was conducted on one day, ie 28 January 2020. Perhaps, more participants would have been recruited over an extended period, say, a week or so, in data collection. Another possible interpretation is platform of recruitment, Credamo, which might not have attracted sufficient participants. The authors have acknowledged this shortcoming. They might be more cautious to draw association of factors in the interpretation of findings in this sample of 1,030 respondents.